# *Aedes* Mosquito Surveillance Using Ovitraps, Sweep Nets, and Biogent Traps in the City of Yaoundé, Cameroon

**DOI:** 10.3390/insects13090793

**Published:** 2022-08-31

**Authors:** Borel Djiappi-Tchamen, Mariette Stella Nana-Ndjangwo, Elysée Nchoutpouen, Idene Makoudjou, Idriss Nasser Ngangue-Siewe, Abdou Talipouo, Marie Paul Audrey Mayi, Parfait Awono-Ambene, Charles Wondji, Timoléon Tchuinkam, Christophe Antonio-Nkondjio

**Affiliations:** 1Vector Borne Diseases Laboratory of the Research Unit Biology and Applied Ecology (VBID-RUBAE), Department of Animal Biology, Faculty of Science, University of Dschang, Dschang P.O. Box 067, Cameroon; 2Institut de Recherche de Yaoundé (IRY), Organisation de Coordination pour la lutte Contre les Endémies en Afrique Centrale (OCEAC), Yaoundé P.O. Box 288, Cameroon; 3Laboratory of Parasitology and Ecology, Department of Animal Physiology and Biology, Faculty of Science, University of Yaoundé I, Yaoundé P.O. Box 337, Cameroon; 4Centre for Research in Infectious Disease (CRID), Yaoundé P.O. Box 13591, Cameroon; 5Laboratory of Biology and Physiology of Animal Organisms, Faculty of Sciences, University of Douala, Douala P.O. Box 24157, Cameroon; 6Vector Biology Liverpool School of Tropical Medicine Pembroke Place, Liverpool L3 5QA, UK

**Keywords:** arboviruses diseases, *Aedes*, sampling methods, rural, peri-urban, urban, Yaoundé, Cameroon

## Abstract

**Simple Summary:**

The emergence/re-emergence of arboviruses diseases including dengue, chikungunya, and yellow fever is of serious public health concern in Cameroon and vector surveillance is a key component of prevention strategies. Entomological surveys were carried out in and around the city of Yaoundé, Cameroon, using different collection methods including ovitraps, Biogent Sentinel traps, and sweep nets to assess *Aedes* species distribution. Results indicated high infestation level of *Aedes* species using ovitraps in the city of Yaoundé. The situation calls for regular surveillance and control of *Aedes* population to prevent the sudden occurrence of outbreaks.

**Abstract:**

Arbovirus diseases represent a significant public health problem in Cameroon and vector surveillance is a key component of prevention strategies. However, there is still not enough evidence of the efficacy of different sampling methods used to monitor *Aedes* mosquito population dynamic in different epidemiological settings. The present study provides data on the evaluation of ovitraps and different adult sampling methods in the city of Yaoundé and its close vicinity. Entomological surveys were carried out from February 2020 to March 2021 in two urban (Obili, Mvan), two peri-urban (Simbock, Ahala), and two rural (Lendom, Elig-essomballa) sites in the city of Yaoundé. The efficacy of three sampling methods, namely ovitraps, Biogent Sentinel trap, and sweep nets, was evaluated. Different ovitrap indices were used to assess the infestation levels across study sites; a general linear model was used to determine if there are statistical differences between positive ovitraps across ecological zones. A total of 16,264 *Aedes* mosquitoes were collected during entomological surveys. Ovitraps provided the highest mosquito abundance (15,323; 91.14%) and the highest species diversity. Of the five *Aedes* species collected, *Aedes* albopictus (59.74%) was the most commonly recorded in both urban and rural settings. Different *Aedes* species were collected in the same ovitrap. The ovitrap positivity index was high in all sites and varied from 58.3% in Obili in the urban area to 86.08% in Lendom in the rural area. The egg density index varied from 6.42 in Mvan (urban site) to 13.70 in Lendom (rural area). Adult sampling methods recorded mostly *Aedes* albopictus. The present study supports high infestation of *Aedes* species in the city of Yaoundé. Ovitraps were highly efficient in detecting *Aedes* distribution across study sites. The situation calls for regular surveillance and control of *Aedes* population to prevent sudden occurrence of outbreaks.

## 1. Introduction

*Aedes*-borne diseases such as dengue, chikungunya, yellow fever, and zika are current public health threats in tropical and sub-tropical regions. Close to 4 billion people across the world are exposed to the risk of transmission of these diseases [1,2]. The prevalence of these arboviral diseases is on the rise across sub-Saharan Africa with important outbreaks registered in major urban settings [3,4,5,6]. In Cameroon, the frequency and magnitude of outbreaks have been on the rise during recent years [6,7,8,9,10,11]. In the city of Yaoundé, the number of dengue cases reported varied from 9.8% (59/603) in 2014 [12] to 12.5% (03/24) in 2018 [13] to 81.81% (90/110) in 2021 [8]. For chikungunya, the number of cases was 15.15% (5/33) in 2006 [14] and 25% (03/12) in 2013 [15]. Studies conducted so far indicated a large distribution of the main arbovirus vectors *Aedes aegypti* and *Aedes albopictus* across the country [16,17,18,19] and the possibility of transovarial transmission of dengue and chikungunya viruses [20,21].

Given the lack of fully protective vaccines or drugs against the majority of arboviruses, vector control through the reduction of *Aedes* breeding sites, environmental management, and improvements in water supply and storage are current practices to reduce the risk of diseases transmission [22]. Vector surveillance could be key for recording *Aedes* species distribution, population density, larval habitats, and for risk prediction. Visual search of larvae and pupae in water containers in and around houses is the main method used for vector surveillance activities. One of the major challenges for surveillance and control of *Aedes* populations using this approach is the wide range of cryptic breeding habitats for *Aedes* mosquitoes and limitation of the index threshold employed to assess transmission risk. Indexes such as the house index (percentage of houses infected by *Aedes* larvae or pupae) and the Breteau index (number of containers found with *Aedes* larvae or pupae for a set of 100 houses inspected) have shown low levels of infestations or negative results in areas with high predominance of *Aedes aegypti* [22]. Moreover, traditional larval indices are inadequate indicators for predicting dengue transmission risk partly because of disparities in the vectorial capacity of *Aedes* mosquitoes across epidemiological settings [23,24]. Given the biological specificities of some *Aedes* species and the cryptic nature of their breeding sites, a combination of methods/tools is currently used to monitor *Aedes* population dynamic in many countries in Africa, America and Asia [25,26,27,28]. These tools could give an accurate picture of *Aedes* species density, distribution, and biting dynamic in a particular environment. These collection methods include ovitraps which are relatively easy and inexpensive to produce and to monitor *Aedes* immature stages [29,30]: Biogents Sentinel trap [31,32], back-pack aspirators [33], and sweep nets [34] to evaluate adults mosquito biting densities. Monitoring adult vector population alongside larval surveys could provide additional information not always captured by larval surveys such as anthropophilic *Aedes* populations and biting dynamic of adults mosquitoes [35,36]. This information could be crucial for planning efficient control strategies.

The study’s main objective was to evaluate the efficacy of different collection methods for the surveillance of *Aedes* species in Yaoundé city and its close vicinity.

## 2. Materials and Methods

### 2.1. Study Sites

The study was conducted in Yaoundé, Cameroon, in both rural and urban settings. Yaoundé is located within the Congo–Guinean phytogeographic zone, and characterized by a typical equatorial climate with four seasons: two rainy seasons (March to June and September to November) and two dry seasons (December to February and July to August) [37]. The city has a population estimated at about 4 million inhabitants and is situated 800 m above sea level (https://populationstat.com/cameroon/yaounde) (accessed on 28 August 2022). Samplings were conducted in six sites: two sites in the urban area (Obili and Mvan), two sites in the peri-urban area (Simbock and Ahala), and two in the rural area (Lendom and Elig-essomballa) (Figure 1).

The rural areas Lendom and Elig-essomballa (separated from each other by about 3000 m) are surrounded by a preserved primary rainforest with dense canopy cover, trees with holes and bamboos. The study sites’ description is presented in detail elsewhere [17].

### 2.2. Study Design

The study was carried out from February 2020 to March 2021. Ovitraps were used to sample immature stages of mosquitoes whereas Biogent Sentinel traps and sweep nets were used for adult collection. An oral consent from each household owner was obtained before mosquito collection around houses. In each study site, 100 ovitraps were placed on trees close to houses to collect *Aedes* immature stages and these traps were monitored for 4 weeks (after every 7 days) in each study site. The traps consisted of small black plastic painted containers with a rounded shape (19.7 cm in height and 14.6 cm wide) covered in its inner side with filter paper soaked in water to collect the eggs laid by *Aedes* females (Figure 2A).

Ovitraps were placed on tree trunk at 1 m above the ground (attached on tree trunk or on any other support) close to human dwellings. Ovitraps were checked weekly (after every 7 days) for the presence of eggs and/or *Aedes* immature stages during 4 weeks in each study site. The trap was considered to be positive when it contained at least one egg or larvae of *Aedes*. After collecting immature stages, ovitraps were thoroughly rinsed and filled with clean water to minimize contamination and a new filter paper was placed. Lost ovitraps were not replaced. Immature stages collected in each ovitrap were brought to the insectary of OCEAC (Organization for the Coordination and fight against the great Endemics diseases in Central Africa) for rearing under controlled conditions (70–80% humidity, 28 ± 1 °C). After emergence, mosquitoes were provided 10% sucrose solution and adults were identified under a binocular magnifying glass using morphological identification keys [38,39].

The Biogent-Sentinel (BG) traps (https://www.bg-sentinel.com/) (accessed on 28 August 2022) containing a Biogent-lure as attractant and sweep nets were used for adults *Aedes* mosquito collection. In the afternoon from 3 to 6 p.m., once per week (for 4 weeks), two BG traps were installed around houses and at the same time. Three collectors using sweep nets also performed *Aedes* adults collection in greenswards. The source of CO_2_ of the Biogent-Sentinel traps consisted of 3 L of water, 700 g of sugar, and 40 g yeast (dry baker’s yeast); all these were added to the 5 L bottle connected to another 0.5 L bottle using pipe. Live mosquitoes collected using these technique were identified and preserved in RNA later (SIGMA Aldrich, Saint Louis, MO, USA) for further molecular analysis. The three collection methods were deployed in the same household area during the survey.

### 2.3. Data Analysis

A general linear model (GLM) was used to determine if there were statistical differences between positive ovitraps across each ecological zone. Moreover, an estimation of ovitrap efficacy was calculated according to the protocols of Gomes (1998) [40] and Focks [22]. The following indices were determined: Ovitrap Positivity Index (OPI): the ratio between the number of ovitraps containing *Aedes* eggs or immature stages and the number of traps examined × 100; Eggs Density Index (EDI) = total number of eggs/total number of positive ovitraps; and the Area Ovitrap Index (AOI): indicating the number of *Aedes* species collected using ovitraps in each study site.

## 3. Results

### 3.1. Aedes Mosquito Distribution following Each Sampling Method across Study Sites

A total of 16,761 mosquitoes belonging to four genera (*Aedes*, *Culex*, *Eretmapodites*, and *Toxorhynchites*) and nine species were collected during the study (Table 1). Out of these, 16,264 were *Aedes*, 445 *Culex*, 44 *Toxorhynchites*, and 8 *Eretmapodites*. Most of the mosquitoes were collected using ovitraps (91.14%), followed by sweep nets (8.26%) and Biogents sentinel traps (0.32%).

*Aedes albopictus* was the most abundant species collected (59.74%), followed by *Ae. simpsoni* (12.57%), *Ae. aegypti* (12.51%), *Ae. contigus* (6.43%), and *Aedes* (*neomelaniconion*) *palpalis* (0.0061%) (Table 2). Some *Aedes* mosquitoes (8.71%) that could not be identified morphologically were grouped under the term *Aedes* spp.

### 3.2. Ovitraps Positivity Indices in Each Ecological Zone

Four *Aedes* species including *Aedes albopictus, Ae. aegypti, Ae. contigus,* and *Ae. simpsoni* were recorded using ovitraps (Table 3). *Aedes albopictus* was the most frequent species collected using these traps in urban (96.33%, n = 3314), peri-urban (65.13%, n = 2490), and rural areas (40.67%, n = 2511). A significant difference (*p* = 0.0193; F-ratio = 3.619) was observed between the proportion of positive ovitraps (ovitraps with *Aedes* larvae or eggs) across study zones. The highest ovitrap positivity rate (83.50 ± 4.79) was recorded in rural settings. High area ovitrap positivity index associated with the presence of either *Ae. albopictus* and/or *Ae. aegypti* was recorded.

### 3.3. Weekly Variation of the OPI and EDI

Apart from the first week after the installation of ovitraps where Ovitrap Positivity Index (OPI) was close to 40% in urban area, the OPI was always above 60% in all sites supporting high infestation rate (Figure 3). The Egg Density Index (EDI) was found to vary more importantly in both rural and urban districts with values varying from 6.44 to 17.45 in rural area and from 4.06 to 10.07 in urban area. This indicator was more stable in peri-urban districts.

The co-occurrence of different *Aedes* species larvae was observed in ovitraps across study sites, with high level of species cohabitating in the same trap noticed in peri-urban and rural settings (Table 4).

### 3.4. Abundance of Adults Aedes Mosquito across Study Sites

Adult individual of the different *Aedes* species collected using sweep net and Biogent sentinel trap varied significantly according to the collection sites. Sweep net and Biogent traps (Table 5) collected high densities in both urban and peri-urban areas. *Aedes albopictus* was the most abundant species collected using these techniques.

## 4. Discussion

High *Aedes* species infestation rate in Yaoundé and its close neighborhood was detected. The Ovitrap Positivity Index was always above 40% in urban district and 60% in peri-urban and rural settings supporting a reproductively active population and frequent occupation of the traps by *Aedes* populations. The egg abundance and density per trap were all high, supporting a high infestation level across study sites. Similar infestation levels have been reported in urban settings in Brazil [41]. The Ovitrap Positivity Index has been reported to be more sensitive to detecting vector presence compared to larval surveys [42]. The density of the *Aedes* mosquito collected using ovitraps (15,323) were three-fold higher than the density collected by larval surveys in breeding containers during one year survey in the same sites [17] revealing the high sensitivity of ovitraps for detecting real *Aedes* infestations density in specific environment. High ovitrap positivity indices were observed around human dwellings, indicating that human activities provide suitable habitats for the maintenance and proliferation of arbovirus vectors, thus increasing the risk of disease transmission. In addition to the greater sensitivity of the traps, they also provided more information on mosquito population abundance and diversity which could be critical for the implementation of effective preventive measures [43,44].

During the present study, ovitraps were exclusively placed outdoors. Studies conducted in Indonesia indicated that ovitraps placed indoors could also be highly effective for vector surveillance activities [45,46]. Both *Aedes aegypti* and *Aedes albopictus* were recorded. *Ae. albopictus* was by far the most frequent species collected. This figure was in accordance with previous studies conducted in the city of Yaoundé [18]. The abundance of *Ae. albopictus* across different ecological zones confirms its high capacity to adapt to different environments and a competitive superiority of the species over *Ae. aegypti* and native species in different type of habitats. The distribution recorded during the present study contrasts with the situation in some countries such as Indonesia and Brazil where *Aedes albopictus* is known to prefer rural areas with high vegetation cover compared to *Aedes aegypti* more prevalent around human habitats in urban settings [47]. The predominance of *Ae. albopictus* in urban and peri-urban areas has been reported in Île de Mayotte [48] and in many countries in Africa [49,50].

High ovitraps indices were recorded for *Ae. albopictus* (0.54 to 0.81) compared to *Ae. aegypti* (0.004 to 0.36). It is considered that areas with ovitrap indices above 0.1% for *Aedes* species may be prone to risk of dengue outbreaks [22,51]. More than two species were commonly recorded per ovitrap which likely supports a high interspecific competition between *Aedes* species since they compete for the same food resources. It is possible that *Aedes* gravid females may practice skip oviposition, which could be part of a dissemination strategy aiming to disperse eggs in order to give more chances to the progeny to grow to the adult stage. This strategy has been reported by many authors [52,53,54].

Many other mosquito species including *Culex*, *Toxorhynchites*, and *Eretmapodites* were also collected alongside *Aedes* species in ovitraps. Although this cohabitation could increase competition between these species, this also suggest that ovitraps could also serve for the surveillance of vectors of other diseases such as filarial vectors (*Culex* species) as previously reported in Italy [55], India [56], and Mexico [29]. High species richness was recorded in rural and peri-urban area compared to urban districts and could result from the diversity of habitats and preserved ecosystems in those areas; similar findings were reported from rural areas in Sri Lanka [57].

In Cameroon, the use of ovitraps for routine surveillance of *Aedes* vector is not widely practiced despite the increasing number of dengue and chikungunya cases across the country [8,9,10,11,58]. The study stresses the need for the promotion of ovitraps for routine surveillance of arboviral vectors across the country.

Surveillance of adults *Aedes* mosquito is also important to monitor the biting dynamic of adult mosquito populations. A high density of adult mosquito was recorded using sweep nets and Biogent sentinel traps. *Ae. albopictus* represented the predominant adult biting species collected using both methods and this observation was similar to studies conducted in Yaoundé [59] and in China [60].

## 5. Conclusions

With the increasing number of dengue and chikungunya cases recorded in Cameroon, it becomes important that timely and accurate entomological data be collected to guide control efforts. Expanding the nationwide use of ovitraps-based monitoring tools in Cameroon could be key for effective monitoring of *Aedes* population dynamic and for predicting possible risk of dengue or chikungunya outbreaks. Involving the community in vector surveillance activities should be promoted also to improve the performance of control interventions.

## Figures and Tables

**Figure 1 insects-13-00793-f001:**
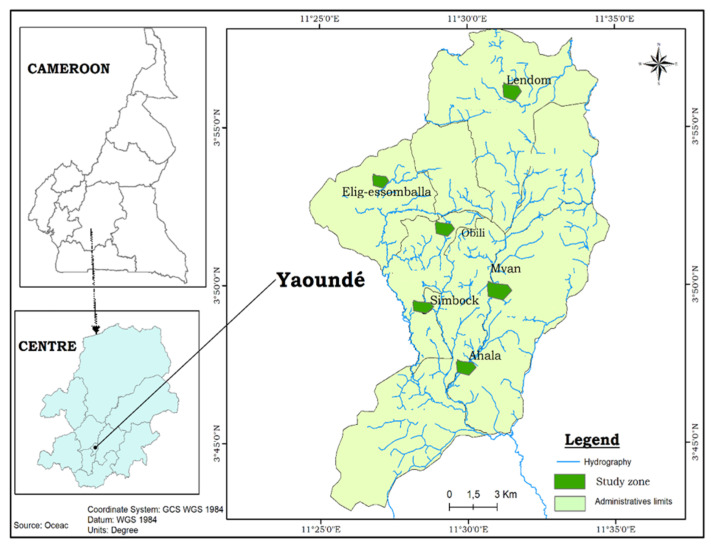
Map of the city of Yaoundé showing the study sites.

**Figure 2 insects-13-00793-f002:**
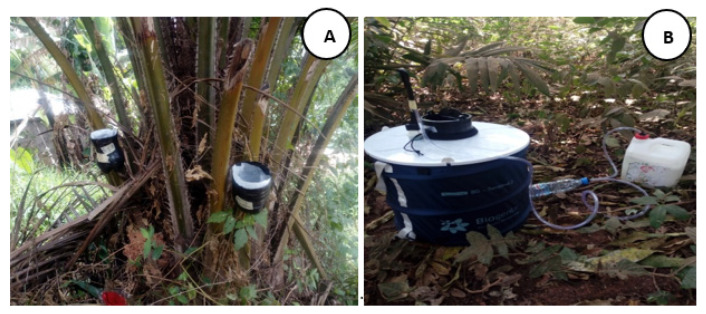
Illustrative image of ovitrap (**A**) and Biogent Sentinel trap (**B**).

**Figure 3 insects-13-00793-f003:**
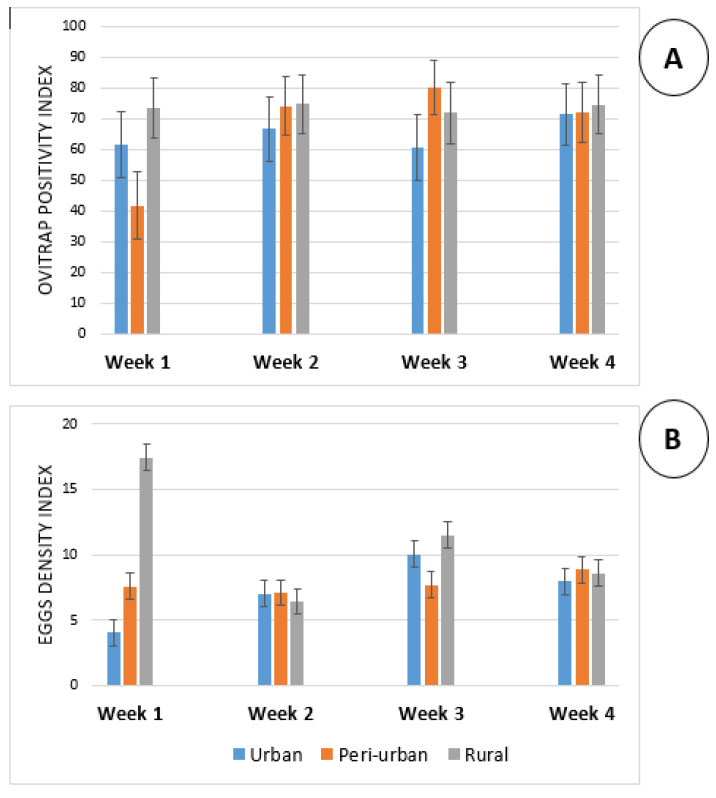
Weekly variation of Ovitraps Positivity Index (**A**) and Eggs Density Index (**B**) around human dwellings.

**Table 1 insects-13-00793-t001:** Abundance of mosquitoes according to sampling techniques and ecological zone.

			Ecological Zones				
	Urban		Peri-Urban		Rural		Total
	Obili	Mvan	Simbock	Ahala	Lendom	Elig-Essomballa	
Ovitraps	1660	1834	3033	1560	1618	5618	15,323
Sweep nets	320	328	169	427	99	41	1384
Biogent-Sentinel trap	11	13	1	24	5	0	54
Total	1991	2175	3203	2011	1722	5659	16,761

**Table 2 insects-13-00793-t002:** Species distribution according to sampling methods.

	Sampling Methods			
Species	Ovitraps	Sweep Net	Biogent Sentinel Trap	Total
*Aedes albopictus*	8315	1348	54	9717
*Aedes aegypti*	2033	3	0	2036
*Aedes contigus*	1047	0	0	1047
*Aedes simpsoni*	2042	3	0	2045
*Aedes* spp.	1418	0	0	1418
*Aedes (neomelaniconion) palpalis*	0	1	0	1
*Culex culiciomayia group*	371	4	0	375
*Toxorhynchites*	43	1	0	44
*Culex moucheti*	13	12	0	25
*Culex lutzia tigripes*	7	1	0	8
*Culex duttoni*	0	1	0	1
*Culex quinquefasciatus*	2	2	0	4
*Culex* spp.	26	6	0	32
*Eretmapodites*	6	2	0	8
Total	15,323	1384	54	16,761

**Table 3 insects-13-00793-t003:** Ovitrap indices in each sampling site.

						Area Ovitraps Index (Species Index)			
Study Areas	N	n	OPI (%)	Number of Eggs	EDI	*Ae. albopictus*	*Ae. aegypti*	*Ae. contigus*	*Ae. simpsoni*
Obili	331	193	58.3 ± 10.80 ^b^	1660	8.60	1658 (0.80)	2 (0.01)	0	0
Mvan	391	277	70.84 ± 9.95 ^a,b^	1780	6.42	1656 (0.81)	26 (0.02)	98 (0.04)	0
Simbock	345	252	73.04 ± 9.72 ^a,b^	2266	8.99	1129 (0.67)	418 (0.31)	390 (0.23)	329 (0.15)
Ahala	381	236	61.94 ± 10.63 ^a,b^	1557	6.59	1361 (0.74)	1 (0.004)	176 (0.11)	19 (0.01)
Lendom	388	334	86.08 ± 7.58 ^a^	4576	13.70	1457 (0.54)	1395 (0.36)	192 (0.09)	1532 (0.41)
Elig-essomballa	391	240	61.38 ± 10.66 ^a,b^	1598	6.65	1054 (0.60)	191 (0.19)	191 (0.21)	162 (0.14)
Total	2227	1532		13,437		8315 (61.88%)	2033 (15.12%)	1047 (7.79%)	2042 (15.19%)

N = total number of ovitraps placed in each site, n = total number of ovitraps with eggs or larvae; OPI = Ovitrap Positivity Index; EDI = Eggs Density Index; (^a,b^) data followed by different letters are significantly different at *p* ˂ 0.05.

**Table 4 insects-13-00793-t004:** *Aedes* species cohabitating in ovitraps.

Species	Urban		Peri-Urban		Rural	
	Obili	Mvan	Simbock	Ahala	Lendom	Elig-Essomballa
*Ae. albopictus + Ae. aegypti*	1.03 (2/193)	2.52% (7/277)	22.22% (56/252)	0.42% (1/236)	14.16% (34/240)	21.25% (71/334)
*Ae. albopictus + Ae. contigus*	0	3.97% (11/277)	17.06% (43/252)	9.32% (22/236)	14.16% (34/240)	8.08% (27/334)
*Ae. albopictus + Ae. simpsoni*	0	0	12.69% (32/252)	1.27% (3/236)	11.25% (27/240)	22.75% (76/334)
*Ae. aegypti + Ae. contigus*	0	0.72% (2/277)	12.69% (32/252)	0.42% (1/236)	5.41% (13/240)	06.88% (23/334)
*Ae. aegypti + Ae. simpsoni*	0	0	9.12% (23/252)	0	5.41% (13/240)	19.46% (65/334)
*Ae. albopictus + Ae. aegypti + Ae. simpsoni*	0	0	9.12% (23/252)	0	4.58% (11/240)	18.56% (62/334)
*Ae. albopictus + Ae. aegypti + Ae. contigus*	0	0	16.66% (42/252)	0.42% (1/236)	5% (12/240)	6.58% (22/334)
*Ae. albopictus + Ae. contigus + Ae. simpsoni*	0	0	7.93% (20/252)	0.84% (2/236)	4.58% (11/240)	7.18% (24/334)
*Ae. aegypti + Ae. contigus + Ae. simpsoni*	0	0	6.34% (16/252)	0	5% (5/240)	6.58% (22/334)

**Table 5 insects-13-00793-t005:** Distribution of adult *Aedes* species collected using sweep nets and Biogent sentinel traps.

		Ecological Zones						
		Urban		Peri-Urban		Rural		Total
Sampling Methods	Species	Obili	Mvan	Simbock	Ahala	Elig-Essomballa	Lendom	
Sweep net	*Ae. albopictus*	320	328	166	425	10	99	1348
	*Ae. aegypti*	0	0	0	0	3	0	3
	*Ae. simpsoni*	0	0	0	0	3	0	3
	*Aedes palpalis*	0	0	0	0	1	0	1
Biogent Sentinel trap	*Ae. albopictus*	11	13	1	24	0	5	54

## Data Availability

The data presented in this study are available in this article.

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
