# Peer review of "Aedes Mosquito Surveillance Using Ovitraps, Sweep Nets, and Biogent Traps in the City of Yaoundé, Cameroon"

_insects, 2022, doi:10.3390/insects13090793_

Round 1
Reviewer 1 Report
I´d like to know epidemiological data in the region of CHIKV, DENV, it will be helpful to better support the role of Aedes albopictus in Cameron and these arboviruses.
My suggestion is also to include at least in comments, some references on transoviarial transmission of DENV in Ae. albopictus eggs. This mechanism plays an important role during inter-epidemic seasons
Author Response
Responses comments Reviewer 1:
Point by point response to reviewers’ comments,
Comment 1: I´d like to know epidemiological data in the region of CHIKV, DENV, it will be helpful to better support the role of Aedes albopictus in Cameroon and these arboviruses.
Response 1: Information on the distribution of both dengue and chikungunya cases added. Pleaese see the first paragraph of the introduction
Comment 2: My suggestion is also to include at least in comments, some references on transoviarial transmission of DENV in Ae. albopictus eggs. This mechanism plays an important role during inter-epidemic seasons.
Response 2: Thank you for this suggestion, the following have been added in the manuscript. Please see the end of the first paragraph of the introduction.
Reviewer 2 Report
Some additional comments provided on attached pdf file.

Author Response
Responses comments Reveiwer 2:
Point by point response to reveiwers’ comments,
Simple Summary:
Comment 1: add ‘’s’’, ‘’such’’ in the sentence
Response 1: corrected
Comment 2: (The present study provides data on the evaluation of ovitraps and different adult sampling methods in the city of Yaoundé and its neighborhood.)
What does this mean, is this part of Yaoundé?
Response 2: sentence rephrased to improve understanding
Abstract:
Comment 3: add ‘’a’’ in the sentence
Response 3: added,
Comment 4: In the sentence ‘’The present study provides data on the evaluation of ovitraps and different adult sampling methods in the city of Yaoundé and its neighbourhood’’, what does this means, is this part of Yaoundé?
Response 4: The term neighborhood was changed to “close vicinity”
Comment 5: Do you mean two?
Response 5: Yes changed
Comment 6: Remove GLM, add a comma on total mosquito number
Response 6: These were corrected as suggested,
Comment 7: ‘’Different Aedes species were found to cohabit in the same ovitrap’’. Do you mean to say, collected in the same trap?
Response 7: Yes, it was changed to collected
Comment 8: Delete the upper ‘’E’’ on word Eggs,
Response 8: done.
Comment 9: ‘’Adults sampling methods recorded mostly Aedes albopictus’’, Duplication, already mentioned this above.
Response 9: not a duplication because it presents Aedes species obtained using adults trapping methods.
Introduction
Comment 10: It would be good to include the risk of diseases in Yaoundé specifically, average number of cases per annum?
Response 10: Information added see the first paragraph of the introduction
Materials and Methods
Study sites
Comment 11: Replace 02 by two
Answer 11: done
Study Design :
Comment 12: Please provide detail if all 03 trap types were deployed in the same household area
Answer 12: Information added each of the three collection methods were deployed in the same household area.
Results:
Comment 13: write the following numbers in full: 04, 09, 08
Answer 13: corrected
Comment 14: please check the full document and insert decimal places consistently
Answer 14: done
Comment 15: Following this expression, ‘’Apart from Day 7’’. Do you mean the first week?
Answer 15: Changed,
Comment 16: I don’t understand the term Day7, Day14, etc on the figure, is this not better to say week1 week2, etc?
Answer 16: it was changed to week 1, week 2 … as requested
Discussion:
Comment 17: Where any of the samples collected evaluated for virus infection rate?
Answer 17: No but samples are preserved for further analysis.
Comment 18: If it is common to find them together, how does this also allow for high interspecific competition?
Answer 18: the information “they compete for the same food resources” was added
Comment 19: How was the diversity calculated (using H/Shannon diversity index?) Not in the method section or the results section. There is a difference between abundance, species richness and diversity.
Answer 19: We change to “diversity” to “species richness”
Conclusion:
Comment 20: It would be nice in the discussion to based the number of cases in relationship to the area where the collection were made.
Answer 20:
The information on the number of cases is not available for the different study sites.
Reviewer 3 Report
In their Manuscript ID: insects-1856707 “ Aedes mosquito surveillance using ovitraps, sweep nets and Biogent traps in the city of Yaoundé, Cameroon”, Borel and co-authors report on one year mosquito surveillance data from a city in Cameroon. The authors apply standard statistical measures to their data to determine the best methodology for vector surveillance. The manuscript is generally well-written; follow scientific convention, with a few relatively minor errors in English language. The conclusions are supported by the data and analyses performed, although it needs revisions before it is acceptable for publication.
Abstract
- The efficacy of three sampling methods including ovitraps, Biogent Sentinel traps and sweep-nets was evaluated.
- Different ovitrap indices…
- … statistical differences between positive ovitraps …
Introduction
- This cites do not follow the way you do it in the rest of the paper: Im et al., 2020; Lutomiah et al., 2016; Simo-Nemg et al., 2019; WHO, 2021.
- The same with: Galani et al., 2021; Nana-Ndjangwo et al., 2022; Simo-Nemg et al., 2018; Tchuandom et al., 2019; WHO, 2021; Yousseu et al., 2018.
- These tools could give an accurate picture of Aedes species density, distribution and biting dynamic in a particular environment.
- … and sweep nets [27].
Materials and Methods
- “The rural areas Lendom and Elig-essomballa (separated from each other by about 1200 m)”: Please check the map, it seems that both areas are separated by a distance longer than 3000 m according to the scale.
- “In each study site, 100 ovitraps were placed on trees close to houses to collect Aedes immature stages and these traps were monitored during 04 weeks (after every 7 days) in each study site”. You previously mentioned that samples were taken from February 2020 to March 2021, does that mean that after four weeks you repeated the whole procedure? Is that correct? Please, specify.
- The trap was considered to be positive when it contained at least one egg or larvae of Aedes
- “… and a new filter paper was placed”: That means that ovitraps were used again? Please, explain it.
- You explained how ovitraps were built, then I suggest to provide a brief explanation about what Biogent-Sentinel traps consist of.
- Data analysis: “… according to the protocols of Gomes (1998) [33] and [15]”. And who [15]? Complete.
Results
- Most of the mosquitoes were collected using ovitraps (15323/16761; 91.14%), … Probably the percentage is enough information. I would remove absolute numbers (15323/16761) since that information can be obtained from Table 1.
- … that could not be identified morphologically were grouped under the term Aedes spp. Aedes spp. (spp without italics).
- High area ovitrap positivity index associated with the presence of either Ae. albopictus and/or Ae. aegypti was recorded.
- Ae. albopictus was the most frequent species collected using these traps in urban …: I suggest not to abbreviate the genus at the beginning of a sentence.
- Apart from Day 7 after the installation of ovitraps where Ovitrap positivity index (OPI) was close to 40% in urban area, the OPI was always above 60% in all sites supporting high infestation rate (Figure 4). I think you mean Figure 3.
- 3.4. Abundance of adult Aedes mosquitoes across study sites.
- Suggestion: Adult individuals of the different Aedes species collected using sweep net and biogent sentinel trap varied significantly according to …
-Table 3: Area Ovitraps index (Species index). I suggest that you explain how you calculate the Species Index, it will be easier to follow.
- You do not mention Table 5 in the text, and Table 4 is mentioned where it should be Table 5. Please check it.
Discussion
- Similar infestation levels have been reported in urban settings in Brazil [34]. The circumstances were the same as here? Environmental conditions were similar?
- (Focks, 2003; Ministry of Health Malay-sia 1997). Check cites.
- High ovitrap positivity indices were recorded around human dwellings, indicating that human activities provide suitable habitats for the maintenance and proliferation of arbovirus vectors
- “… as previously reported in Italy [48], India [49] and Florida, USA [22]”. Paper 22 refers to a surveillance carried out in Mexico, please check the bibliography.
- High species diversity was recorded in rural and peri-urban area compared to urban districts …
- “A high density of adult mosquito was recorded using sweep net and biogents sentinel traps”. I think that this should be discussed deeply, since you mention in the Title this methodology the reader would expect that.
Author Response
Responses comments Reveiwer 3:
Point by point response to reveiwers’ comments,
Abstract:
Comment 1:
- The efficacy of three sampling methods including ovitraps, Biogent Sentinel traps and sweep-nets was evaluated.
- Different ovitrap indices…
- … statistical differences between positive ovitraps …
Answer 1: Thank you dear reviewer for these remarks, the following have been corrected; please see below:
The efficacy of three sampling methods including ovitraps, Biogent Sentinel trap and sweep-nets was evaluated. Different ovitrap indices were used to assess infestation levels across study sites; a general linear model was used to determine if there is statistical differences between positive ovitraps.
Introduction
Comment 2:
- This cites do not follow the way you do it in the rest of the paper: Im et al., 2020; Lutomiah et al., 2016; Simo-Nemg et al., 2019; WHO, 2021.
- The same with: Galani et al., 2021; Nana-Ndjangwo et al., 2022; Simo-Nemg et al., 2018; Tchuandom et al., 2019; WHO, 2021; Yousseu et al., 2018.
- These tools could give an accurate picture of Aedes species density, distribution and biting dynamic in a particular environment.
- … and sweep nets [27].
Answer 2:
Thanks for this remark, the references were put in the format of the journal
Materials and Methods
Comment 3:
- “The rural areas Lendom and Elig-essomballa (separated from each other by about 1200 m)”: Please check the map, it seems that both areas are separated by a distance longer than 3000 m according to the scale.
Answer 3: changes done
Comment 4:
- “In each study site, 100 ovitraps were placed on trees close to houses to collect Aedes immature stages and these traps were monitored during 04 weeks (after every 7 days) in each study site”. You previously mentioned that samples were taken from February 2020 to March 2021, does that mean that after four weeks you repeated the whole procedure? Is that correct? Please, specify.
Answer 4: Yes the whole procedure was repeated in each study site.
Comment 5:
- The trap was considered to be positive when it contained at least one egg or larvae of Aedes
Answer 5: Yes thank you
Comment 6:
- “… and a new filter paper was placed”: That means that ovitraps were used again? Please, explain it.
Answer 6: Yes, after that ovitraps were verified for Aedes immatures stages, its content was removed, rinsed and a new filter paper was added in all the ovitraps for further verifications on the next 7 days.
Comment 7:
- You explained how ovitraps were built, then I suggest to provide a brief explanation about what Biogent-Sentinel traps consist of.
Answer 7: the web site for biogent sentinel trap added
Comment 8:
- Data analysis: “… according to the protocols of Gomes (1998) [33] and [15]”. And who [15]? Complete.
Answer 8: Thanks, the following was corrected as follows:
Also, an estimation of ovitrap efficacy was calculated according to the protocols of Gomes (1998) [40] and Focks [22].
Results
Comment 9:
- Most of the mosquitoes were collected using ovitraps (15323/16761; 91.14%), … Probably the percentage is enough information. I would remove absolute numbers (15323/16761) since that information can be obtained from Table 1.
Answer 9: Information removed as requested
Comment 10:
- … that could not be identified morphologically were grouped under the term Aedes spp. Aedes spp. (spp without italics).
Answer 10: corrected
Comment 11:
- High area ovitrap positivity index associated with the presence of either Ae. albopictus and/or Ae. aegypti was recorded.
Answer 11: corrected
Comment 12:
- Ae. albopictus was the most frequent species collected using these traps in urban …: I suggest not to abbreviate the genus at the beginning of a sentence.
Answer 12: corrected
Comment 13:
- Apart from Day 7 after the installation of ovitraps where Ovitrap positivity index (OPI) was close to 40% in urban area, the OPI was always above 60% in all sites supporting high infestation rate (Figure 4). I think you mean Figure 3.
Answer 13: corrected
- 3.4. Abundance of adult Aedes mosquitoes across study sites.
Comment 14:
- Suggestion: Adult individuals of the different Aedes species collected using sweep net and biogent sentinel trap varied significantly according to …
Answer 14: corrected
Comment 15:
-Table 3: Area Ovitraps index (Species index). I suggest that you explain how you calculate the Species Index, it will be easier to follow.
Answer 15: Information added in the Method section
the Area Ovitrap Index (AOI)= the abundance (number) of Aedes species collected using ovitraps in each study site/total number of Aedes collected
Comment 16:
- You do not mention Table 5 in the text, and Table 4 is mentioned where it should be Table 5. Please check it.
Answer 16: This has been modified accordingly
Discussion
Comment 17:
- Similar infestation levels have been reported in urban settings in Brazil [34]. The circumstances were the same as here? Environmental conditions were similar?
Answer 17: Not totally but here the comparisons were mainly based eggs abundance among ovitraps.
Comment 18:
- (Focks, 2003; Ministry of Health Malay-sia 1997). Check cites.
Answer 18: Corrected
- High ovitrap positivity indices were recorded around human dwellings, indicating that human activities provide suitable habitats for the maintenance and proliferation of arbovirus vectors
- “… as previously reported in Italy [48], India [49] and Florida, USA [22]”. Paper 22 refers to a surveillance carried out in Mexico, please check the bibliography.
Answer 19: Thanks for this remark, this was corrected accordingly
Comment 19:
- High species diversity was recorded in rural and peri-urban area compared to urban districts …
Answer 19: corrected
Comment 20:
- “A high density of adult mosquito was recorded using sweep net and biogents sentinel traps”. I think that this should be discussed deeply, since you mention in the Title this methodology the reader would expect that.
Answer 20: Dear reviewer, results of these activities concerning mosquito abundance and diversity were not very high and we made our possible to explain these findings.
Round 2
Reviewer 3 Report
Comment 9 (from previous review):
- Most of the mosquitoes were collected using ovitraps (15323/16761; 91.14%), … Probably the percentage is enough information. I would remove absolute numbers (15323/16761) since that information can be obtained from Table 1.
Answer 9: Information removed as requested
- I suggested to remove the absolute numbers , not the percetages. Absolute numbers are already in Table 1 but the percentages are not (e.g: 91.14%).
- Point 3.2- 40.67% n=2,511 Include coma or semicolon between numbers (40.67%, n=2,511)
Author Response
Responses comments Reveiwer 3:
Point by point response to reveiwers’ comments,
Comment 1:
suggested to remove the absolute numbers , not the percetages. Absolute numbers are already in Table 1 but the percentages are not (e.g: 91.14%).
Response 1:
Thanks, information modified as requested,
Please see the modification in yellow in the manuscript
Comment 2:
- Point 3.2- 40.67% n=2,511 Include coma or semicolon between numbers (40.67%, n=2,511)
Response 2:
Thanks, this was added, please see the modification in yellow in the manuscript